# Antimicrobial resistance dissemination via horizontal gene transfer is constrained in stratified waters
**Máté Vass** [1,2,3] ✉**, Anna Abramova**[1,2] **& Johan Bengtsson-Palme**[1,2,4]

Aquatic ecosystems are major reservoirs of antibiotic resistance genes (ARGs) and hubs for microbial interactions that can facilitate their spread through horizontal gene transfer (HGT). While mobile genetic elements (MGEs), including plasmids and viruses, are recognized as important drivers of ARG mobility, the extent to which water column stratification constrains their vertical dissemination remains unresolved. Here, we analysed depth-resolved metagenomic data from stratified freshwater and marine systems to assess the role of HGT in ARG spread. We found that ARG diversity is consistently lower in marine than freshwater environments and that only a small fraction of ARGs is mobilized by plasmids and viruses. Importantly, we detected no evidence for recent HGT-mediated dissemination of ARGs across depth layers, despite genetic compatibility among co-occurring bacteria. Instead, ARGs appear largely confined to lineage-specific inheritance and within-layer persistence. These findings suggest that stratification acts as a barrier, limiting vertical ARG transfer while promoting within-layer accumulation. Given projections of intensified and prolonged stratification under climate change, our results imply reduced vertical connectivity of ARGs in aquatic environments, with potential consequences of further mitigation in its dynamics by water stratification.

Aquatic ecosystems are important reservoirs for antibiotic resistance genes (ARGs) and act as sinks for a multitude of bacteria from anthropogenic sources that occasionally harbour high levels of clinically-relevant resistance genes[1]. These genes can be exchanged with aquatic bacteria through the process of horizontal gene transfer (HGT) which plays a crucial role in microbial evolution and adaptation. HGT is driven by various mobile genetic elements (MGEs) that can be transferred within a genome or between genomes, the latter thanks to plasmids and viruses[2]. Consequently, diverse interactions between bacteria and MGEs can facilitate the spread of antimicrobial resistance (AMR) across diverse microbial communities[3]. Both bacteriophages and giant viruses play a critical role in the acquisition and distribution of ARGs, either through direct transfer or by acting as gene reservoirs[4,5]. Nevertheless, the gene transfer frequencies within and between microbial communities vary[6] and are regulated by various factors[7,8].

Beyond biological processes, physical processes such as the mixing of water masses (or lack thereof) affect microbial dispersal and species interactions, hence, indirectly influencing gene exchanges in aquatic environments. Vertical structuring of water masses, due to temperature or density-driven stratification, can create distinct microbial niches. Stratification is expected to strengthen and prolong under climate change[9–11], leading to altered mixing regimes, in turn shaping the HGT dynamics and resistome at distinct water depths[12–14]. Although such physical phenomena may have important effects on the spatial spread of AMR in aquatic ecosystems, their extent is unclear. Similarly, it is not known whether HGT can act as a mixing force to curb microbial dispersal barriers during stratification and facilitate the spread of AMR.

By utilising metagenomic sequencing data from depth-resolved field studies, our goal was to elucidate the mechanisms underlying the vertical dissemination (or lack thereof) of ARGs via HGT in a diverse set of stratified marine and freshwater environments (Supplementary Fig. 1). Understanding these processes is critical for predicting the persistence and spread of resistance determinants, particularly in the presence of density layers that act as barriers in their dispersal. Guided by various ARG databases to cover both well-known and latent, i.e., ARGs that are not (yet) encountered in clinical pathogens, our findings reveal a generally lower level of ARG diversity in marine than in freshwater environments, and further that

¹Division of Systems and Synthetic Biology, Department of Life Sciences, Science for Life Laboratory, Chalmers University of Technology, Gothenburg, Sweden. ²Centre for Antibiotic Resistance Research (CARe) in Gothenburg, Gothenburg, Sweden. ³Division of Microbial Ecology, Department of Aquatic Sciences and Assessment, Swedish University of Agricultural Sciences, Uppsala, Sweden. ⁴Department of Infectious Diseases, Institute of Biomedicine, The Sahlgrenska Academy, University of Gothenburg, Gothenburg, Sweden. ✉e-mail: mate.vass@slu.se

dissemination of ARGs via HGT events is not evident across depths. We also demonstrate that only a small fraction of the identified ARGs is mobilised by plasmids and viruses, and these genes do not spread extensively across water columns. Our study, thus, provides a comprehensive assessment of the spatial spread of ARGs in both marine and freshwater waters and suggests a weakened role of HGT-mediated spread of AMR and a potential mitigation in its dynamics by water stratification.

## Results

### Horizontal gene transfers in water columns are more prevalent in freshwater than marine water

By using the computational tool MetaCHIP[15] to estimate putative horizontal gene transfers across depths, we identified 4,562 and 100 HGT events in each sampled water column in freshwater and marine environments (Supplementary data 6) among 7,865 and 1,489 metagenome-assembled genomes, respectively (Supplementary data 7 and 8). A small Swiss lake, Le Loclat, and several marine sites in the Southern Hemisphere lacked any predicted gene transfers along water columns (Fig. 1). To enable comparison of HGT frequency across environments with variable taxa diversity, we introduce HGT activity, which refers to the proportion of HGT-mediated unique genes in relation to the total number of genes within a sampling location. We observed significantly greater HGT activity in freshwater than in marine communities (Wilcoxon rank-sum test, $p < 0.001$, $r = 0.48$) (Fig. 1). While this effect size is commonly interpreted as a moderate effect, we caution that this estimate may be influenced by the non-normal distribution of the data (Supplementary Fig. 2). Therefore, we rely primarily on the direction of the median difference rather than the conventional r threshold alone. Pairwise comparisons of HGT activity within freshwater environments revealed no statistically significant differences after Holm-Bonferroni correction for multiple testing (all adjusted $p > 0.05$), however, this should be interpreted with caution due to the small sample sizes in pond ($n = 5$) and reservoir ($n = 1$) categories (Supplementary Fig. 3).

Vertical sampling differed in depth resolution; in the shallow freshwater environments, samples were taken from narrower layers, whereas in marine environments the sampled layers spanned over larger depth intervals. This may have influenced the estimated HGT activity due to the larger depth intervals in marine environments (Supplementary Fig. 4). We observed a negative correlation between median depth interval and HGT activity (Spearman rho = −0.59, $p < 0.001$), suggesting that coarser vertical sampling in marine environments reduced the detectability of HGT events. To statistically control for this confounding factor, we performed an ANCOVA including median depth interval as a covariate and testing for interaction with environment. Environment remained a significant predictor of HGT activity ($p = 0.021$), while depth interval and the interaction were not significant ($p = 0.25$), indicating that the observed environmental difference in HGT activity is not fully explained by differences in sampling resolution.

Gammaproteobacteria ($n_{ORF} = 1825$) and Chlorobia ($n_{ORF} = 1429$) in freshwater environments, and Gammaproteobacteria ($n_{ORF} = 70$) and Acidimicrobiia ($n_{ORF} = 12$) in marine environments were most involved in horizontal gene transfers (Fig. 2). Some taxonomic classes exhibited a biased role distribution (i.e., donor vs. recipient) in freshwater bacteria (Fisher's exact test: $p = 0.001$), whereas such bias was not observed among marine bacteria (Fisher's exact test: $p = 0.587$). For example, in freshwaters, Chlorobia was more prone to be HGT-recipient ($n_{ORF} = 785$) than donor ($n_{ORF} = 644$). Interestingly, a clear directional pattern emerged in ponds, as Chlorobia metagenome-assembled genomes (MAGs) in deeper layers acted as HGT-donors and exchanged genes within its populations in upper water layers (HGT-recipients) (Fig. 2a). HGTs were not evenly distributed across COG (Clusters of Orthologous Groups) functional categories in either environment (Marine: $\chi^2 = 41.8$, df = 18, $p = 0.0012$; Freshwater: $\chi^2 = 1070.6$, df = 23, $p < 0.001$). The greatest number of HGT-mediated genes were assigned to COG categories 'J' (Translation, ribosomal structure and biogenesis) and 'E' (Amino acid transport and metabolism) in both freshwater and

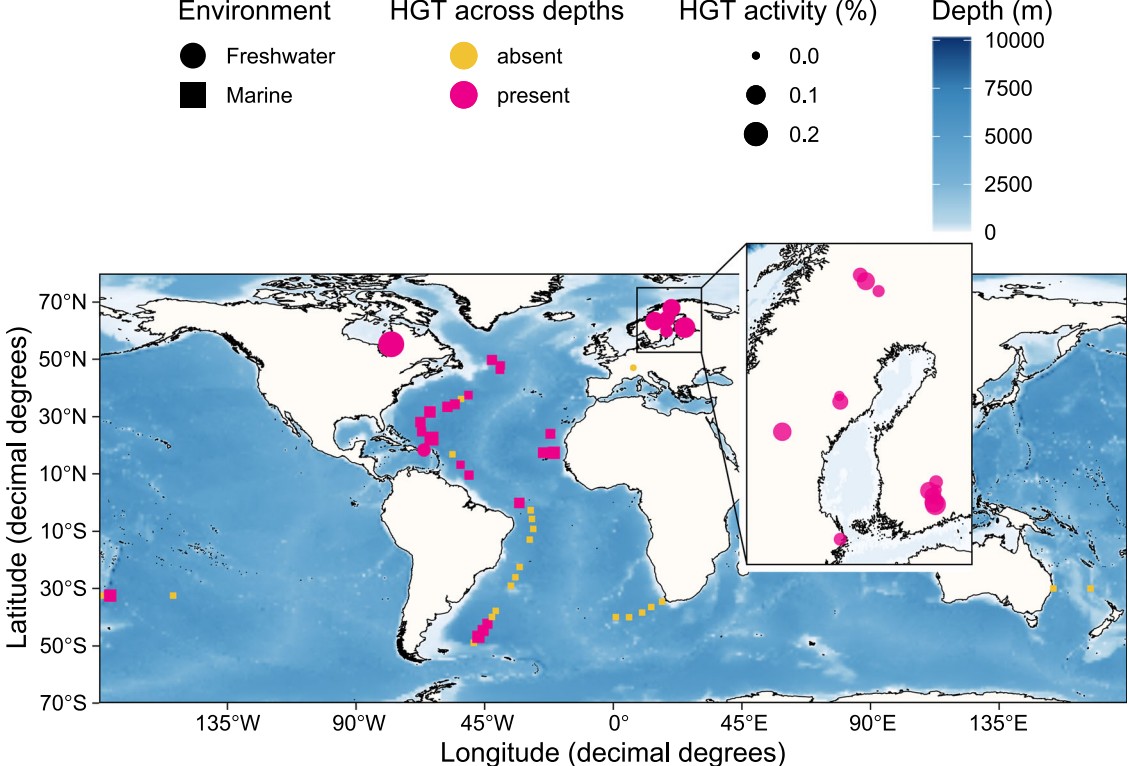

**Fig. 1 | Frequency of estimated horizontal gene transfer (HGT) events.** HGT activity [%], that is, the proportion of HGT-mediated unique genes, expressed as a percentage of the total number of genes (open-reading frames or ORFs) within each sampling location (with at least three sampling depths) in freshwater ($n = 27$) and marine ($n = 41$) environments.

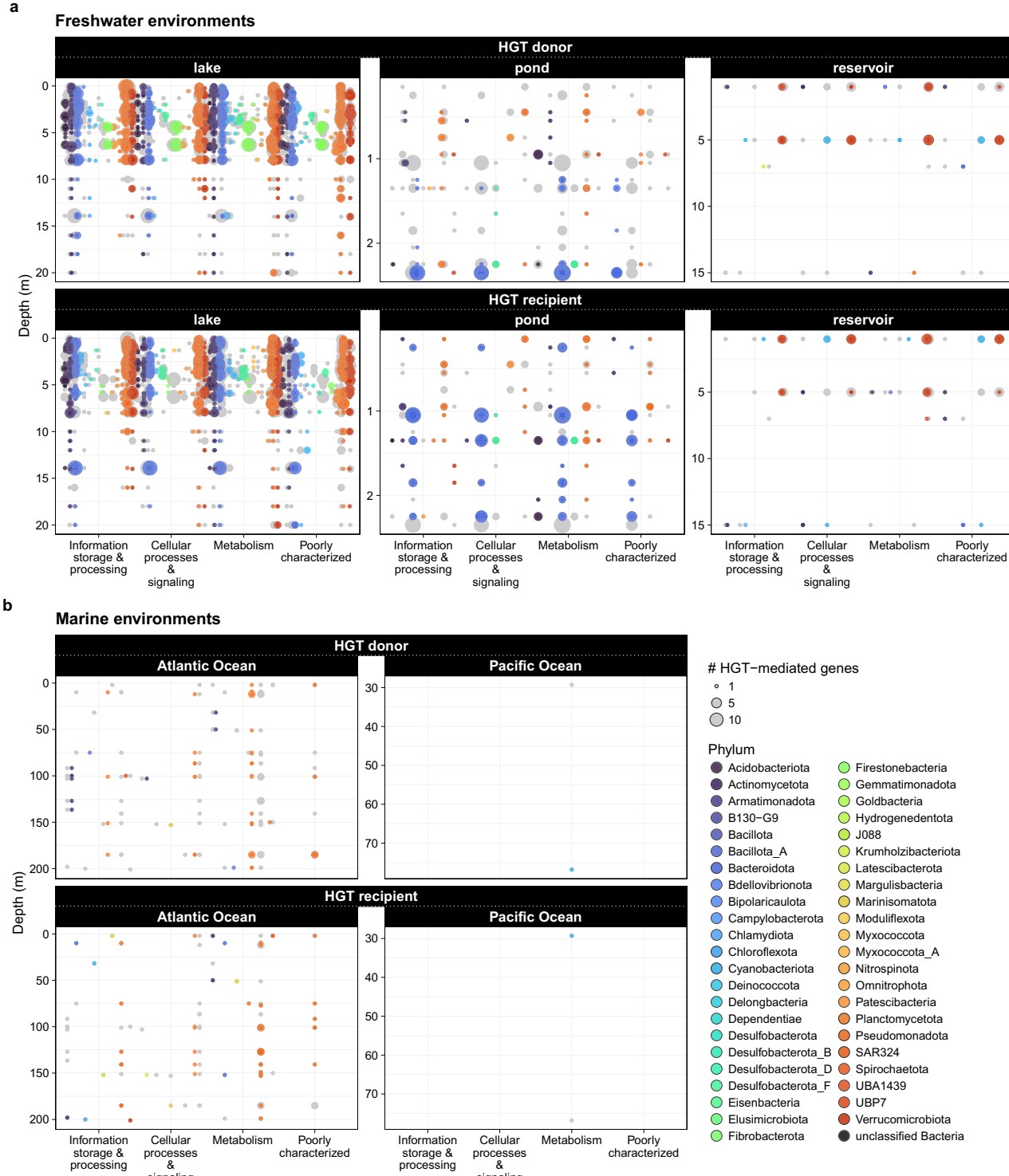

**Fig. 2 | Distribution of horizontally transferred genes.** Direction (i.e., donor or recipient) of horizontally transferred gene flows, and their assigned phylum-level taxonomy (based on GTDB[46]) and functions in various (**a**) freshwater, such as lake ($n = 21$), pond ($n = 5$) and reservoir ($n = 1$) and (**b**) marine (Atlantic Ocean: $n = 36$, Pacific Ocean: $n = 5$) environments.

marine environments, and 'J' was significantly more involved than other gene categories, regardless of the environment (Marine: residual = 5.3; Freshwater: residual = 14.2). Despite the frequent presence of genes associated with defence mechanisms (COG category 'V'), as proxy for stressors (see Supplementary Fig. 5), these genes were generally underrepresented in gene exchanges (Marine: residual = 0.2, n.s.; Freshwater: residual = −3.9) (Supplementary data 9).

**Lack of HGT-mediated antibiotic resistance genes across depths**
All predicted open reading frames (ORFs) were mapped against three distinct ARG databases to cover both pathogenic, non-pathogenic/non-culturable and computationally predicted (latent) ARGs. This analysis resulted in variable distributions of ARGs concentrated mainly in the upper water layers (Fig. 3, Supplementary data 10), where MAGs have been more abundant (Supplementary Fig. 6). Notably, freshwater MAGs carried

greater number of ARGs than marine ones (Supplementary data 11). The vertical distribution of unique ARGs showed differing patterns depending on the type of environment and the reference database applied (Supplementary Fig. 7). In lakes, the ARG richness increased with water depth (Spearman: $r_{CARD} = 0.34$, $p = 0.005$; $r_{ResFinderFG2.0} = 0.40$, $p = 0.007$; $r_{LatentARGs} = 0.50$, $p = 0.002$), while the clinically-relevant ARGs decreased in ponds (Spearman: $r_{CARD} = -0.57$, $p = 0.034$). Other freshwater sites (i.e., reservoir) and marine environments did not result in spatially different ARG distributions (Spearman: $p > 0.05$). Six resistance genes from Gammaproteobacteria MAGs in marine environments were identified by more than one reference database (Supplementary data 12), whereas no overlapping genes were found in freshwater environments. Among those six overlaps in marine MAGs, separate ARGs mapped to CARD[16] or latent ARG[17] databases were also detected by ResFinderFG2.0[18], suggesting a slightly broader coverage of ResFinderFG2.0. Five of the overlapping ARGs conferred resistance to beta-lactams and synthetic antibiotics (i.e., sulfonamides). One case, however, showed a discrepancy as it was annotated as a gene conferring resistance to the MLS (macrolide, lincosamide and streptogramin) super-group by the latent ARG database, but to diaminopyrimidine using the ResFinderFG2.0 database. Notably, no ARG was simultaneously detected by all three applied reference databases.

The total number of ARGs in each aquatic environment differed (Freshwater: lakes = 132, ponds = 21, reservoir = 25; Marine: Atlantic Ocean = 32, Pacific Ocean = 5) (Supplementary data 10). Surprisingly, none of the identified ARGs were involved in the estimated HGT events (Supplementary data 11). Most ARGs in lakes, carried by Pseudomonadota ($n_{ARG} = 46$), Bacteroidota ($n_{ARG} = 37$) and Verrucomicrobiota ($n_{ARG} = 32$), conferring resistance to cycloserine-like ($n = 20$), diaminopyrimidine ($n = 19$), penam ($n = 18$), glycopeptide ($n = 17$) and tetracycline ($n = 17$) antibiotics. Possibly due to the low number of sampling locations, the numbers of unique ARGs, detected in Verrucomicrobiota ($n_{ARG} = 13$) in ponds and Actinomycetota ($n_{ARG} = 8$) in reservoir, conferring resistance mainly to glycopeptide (n = 10 in both environments), were lower than those found in lakes. 38 unique ARGs, mainly harboured by Pseudomonadota ($n_{ARG} = 25$), were found in the sampled transects in the Atlantic Ocean, while only five in the transect in the Pacific Ocean, near Australia and New Zealand (i.e., GP13 GEOTRACES section) and mainly carried by three Cyanobacteriota MAGs. In both oceans, glycopeptide resistance dominated in marine environments ($n_{AtlanticOcean} = 11$, $n_{PacificOcean} = 3$) (Fig. 3).

A total of 34 and 11 ARGs were shared across at least two taxonomic classes in freshwater and marine environments, respectively. In freshwaters, the glycopeptide-resistance *vanT* (*vanG* cluster) gene was the most widely distributed, occurring in 62 classes among 27 phyla, tetracycline- and fluoroquinolone-resistance *adeF* gene was found in 31 classes spanning 14 phyla, while *vanW* (*vanI* cluster) and *vanY* (*vanB* cluster) genes (both glycopeptide-resistance genes) occurred in 25 classes (9 phyla) and 16 classes (10 phyla), respectively. In marine environments, *vanT* (*vanG* cluster) was also the most widely distributed ARG, detected in 12 classes across 11 phyla. Additionally, another vancomycin-resistance gene, *vanY* (*vanB* cluster), and the *qacG* gene, conferring resistance to a broad-spectrum antiseptics and disinfectants, were each identified in five classes from four phyla (Fig. 3).

### Limited vertical spread of mobile genetic elements with resistance genes

Since no HGT-mediated ARG dissemination across water column was evident by the applied community-level HGT estimation tool, we wished to validate our finding from a different perspective and evaluate the presence of ARGs physically encoded on contigs classified as mobile genetic elements (MGE-ARG), independently of MetaCHIP. The presence of such MGE-ARG indicates potential mobility but does not imply recent HGT. We recovered and validated the presence of various viruses (contig lengths ≥ 3,000 bp) with the dominance of Uroviricota contigs (Supplementary Figs. 8, 9) and plasmids (contig lengths: Marine: 221–87,517 bp, Freshwater:

200–357,022 bp) (Supplementary Figs. 10, 11). Most plasmid contigs (Marine: 91.1%, Freshwater: 78.8%) lacked conjugation gene that would otherwise aid ARG dissemination via conjugation. Nonetheless, we wanted to confirm whether plasmids, circulating outside of cells, and viruses could have carried ARGs. Therefore, MGE-ARG contigs were mapped to ARG-carrying bacterial genomes (hereafter MAG-ARGs) (see Methods for details). We assumed that a successful match would indicate that the same ARG sequence is physically encoded on a plasmid or viral contig and also present in a bacterial genome, suggesting potential mobilization of the MGE-ARG. While this demonstrates physical association of ARGs with MGEs, it does not prove that the MGE and host genome were physically linked within the same cell at the time of sampling, nor does it imply recent HGT. Interestingly, only a small fraction of MAG-ARGs (Freshwater: 6.25%, Marine: 14.71%) was mobilised (Fig. 4a, d). Moreover, ARG-carriage by MGEs was mainly limited to one (or maximum two) unique ARGs per sampling site (Fig. 4b, e), indicating a low level of mobility of ARGs by plasmids or viruses in our selected data. We also found evidence for a limited vertical distribution of MAG-ARGs by MGE-ARGs in the water columns (Fig. 4c, f). Here, dots along the 1:1 line indicate that MAG-ARGs at a given depth were found in MGEs at the same depth, reflecting limited vertical dissemination of ARGs via plasmids or viruses across the stratified water column. When analysed by MGE type, freshwater plasmid-associated ARGs showed a highly constrained distribution with a small but significant spread towards shallower depths. This was quantified as the paired difference in depth between MGE-ARG occurrence and the reference depth of the MAG-ARG, where negative values indicate shallower positions in the water column (Wilcoxon signed-rank test: median shift: 0, mean shift: $-0.45$, sd: 0.89, $p < 0.001$). This indicates that while most plasmid-associated ARGs occur at same depths as their hosts, a slight spread (shift) toward shallower depths is present. In contrast, virus-associated ARGs in freshwater exhibited larger variance in depth differences (mean shift: $-1.16$, sd: 4.09), but this spread was not statistically significant (Wilcoxon signed-rank test: median difference: $-0.55$, $p = 0.078$). This suggests sporadic vertical displacement events rather than consistent or directional vertical dissemination. This was mainly attributed to Lake Erken, where phages carried the same glycopeptide-resistance *vanY* (*vanB* cluster) gene that was present in a *Polynucleobacter* host (*Polynucleobacter* sp002292975; MAG: Erken-D6_32_00868) (Fig. 4c, Supplementary data 13). In marine environments, sample sizes for MGE-ARGs were limited, particularly for viruses, precluding robust statistical inference. However, the median depth difference between MGE-ARG occurrence and MAG-ARG was centered at zero, indicating no systematic vertical dissemination of marine plasmid-associated ARGs (Fig. 4f).

Utilising the protein composition and association to viral contigs (≥ 3,000 bp) predicted by a language model (PhaTYP module[19] of PhaBOX[20]), we were able to identify phages with virulent (lytic) and temperate (lysogenic) lifestyles from freshwater samples (Fig. 5, Supplementary data 13). This revealed that the *vanY* gene (*vanB* cluster) in Lake Erken was disseminated by a virulent phage. Two additional unique ARGs were also carried by phages: the *rsmA* (fluoroquinolone, diaminopyrimidine, phenicol) gene was carried by temperate (pro)phages, while glycopeptide-resistance was apparently mediated by virulent phages, carrying *vanW* (*vanI* cluster). All identified phages were classified as members of the Caudoviricetes class (Uroviricota phylum).

### Discussion

In this study, we investigated the vertical dissemination of antibiotic resistance genes via horizontal gene transfer in stratified aquatic ecosystems. Despite widespread evidence of gene exchange across microbial communities and depths, our analysis of over nine thousand genomes reveals no clear signal of recent ARG dissemination between water layers. These findings suggest that stratification creates ecological barriers to ARG mobility, constraining their vertical spread in aquatic environments.

We observed substantially higher HGT activity in freshwaters compared with marine sites. The difference is biologically plausible: marine

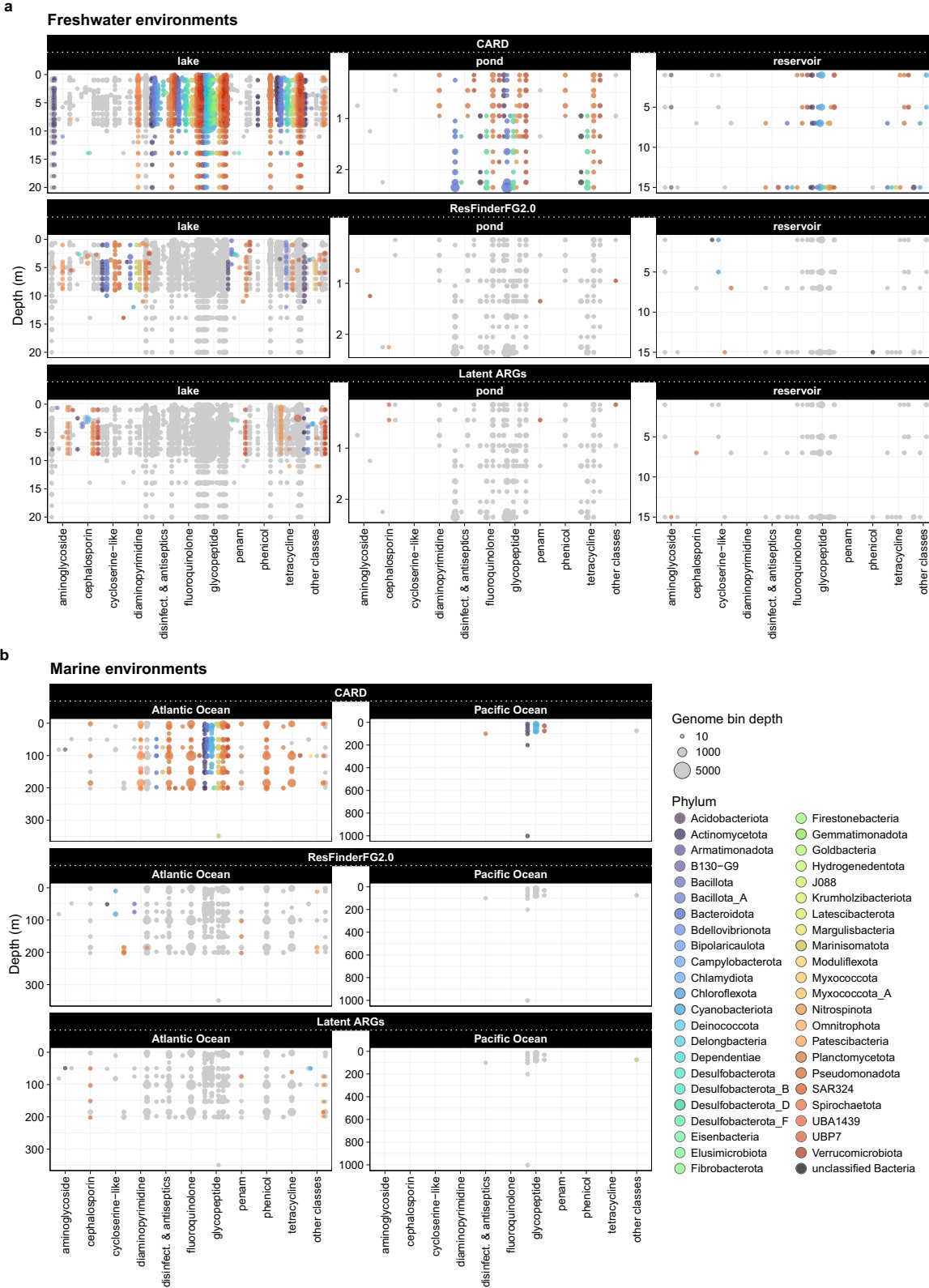

**Fig. 3 | Vertical distribution of the identified antibiotic resistance genes.** Resistance genes conferring resistance to the ten most common antibiotic classes and their associated genomes classified at phylum-level taxonomy (based on GTDB[46]) in various (**a**) freshwater, such as lake (*n* = 21), pond (*n* = 5) and reservoir (*n* = 1), and (**b**) marine (Atlantic Ocean: *n* = 36, Pacific Ocean: *n* = 5) sites. Comprehensive Antibiotic Resistance Database (CARD)[16], the ResFinderFG2.0[18] and latent ARGs, a database compiled by ref. 17, were used as reference databases.

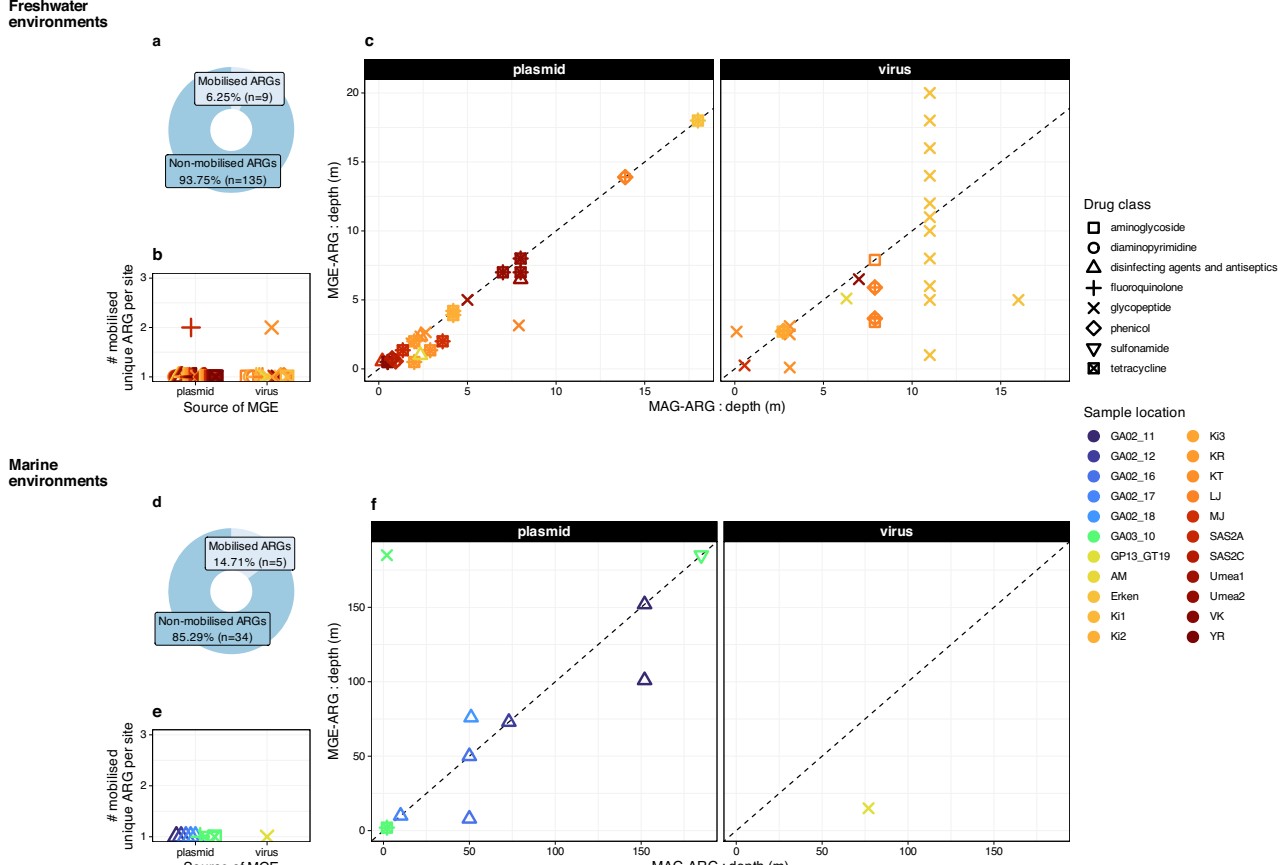

**Fig. 4 | ARG-carrying mobile genetic elements (MGE-ARG). a**, **d** Mobility of antibiotic resistance genes (ARGs) conferring resistance to various drug classes, (**b**, **e**) their occurrences on mobile genetic elements such as plasmids and viruses (≥ 3,000 bp), as well as (**c**, **f**) their mapped presence in genome bins (MAG-ARGs) along water depths. For details on Sample locations, see, Supplementary data 16, 17.

environments, particularly oligotrophic waters, have lower microbial biomass and nutrient availability, which likely reduces both the frequency of HGT events and the probability of detecting them. Sampling resolution along depth may also influence our estimates, with freshwater sites having higher coverage across multiple depth intervals. Therefore, while absolute HGT counts should be interpreted cautiously, the observed pattern reliably reflects higher relative HGT activity in freshwater communities.

Previous studies have explored the vertical distribution of ARGs in aquatic systems, revealing context-dependent patterns that are influenced by environmental conditions. For instance, in estuarine and marine waters, surface layers often show higher ARG abundance, largely driven by anthropogenic inputs and selective pressure in upper, nutrient-rich layers, whereas more stable bottom layers are thought to experience lower selective pressure[13]. The spatial distribution of tetracycline resistance gene *tetA*, as particle-associated gene in Lake Maggiore reflects a predominantly unidirectional sinking from the surface waters to the sediment[12]. In a stratified Chinese reservoir, stratification strongly shaped ARG communities, with particle-attached bacteria harbouring more ARGs and exhibiting higher HGT rates due to close proximity and competitive interactions, whereas mixed conditions promoted nutrient cycling and reduced ARG maintenance[14]. While these studies provided valuable snapshots of ARG dynamics, they often focused on single genes, single sites, or abundance metrics without explicitly evaluating cross-layer gene transfer at community level in a diverse set of aquatic ecosystems. In contrast, our analysis integrates metagenome-assembled genomes from diverse freshwater and marine systems, mapping predicted genes against three complementary ARG databases to capture both well-known (established) and computationally predicted (presumably latent) resistance determinants[17]. We

observed that ARG richness generally increased with depth in freshwater systems, while clinically relevant ARGs decreased in ponds, and no clear vertical pattern was detected in marine sites, in contrast to the finding of[21] which suggested increased ARG diversity with depth. Freshwater MAGs carried a greater number of unique ARGs than marine ones, with distribution patterns differing across environments and reference databases.

Mobile genetic elements and giant viruses are recognized as important drivers of ARG transfer[4,5]. In aquatic ecosystems, phage life cycles are strongly shaped by environmental conditions, with lytic (virulent) activity dominating during productive (eutrophic) periods and lysogeny (temperate) prevailing under oligotrophic conditions[22,23]. Even within a waterbody, stratification can create distinct conditions that would pave the way for phages with altered lifestyle[24]. Such dynamics can amplify or suppress ARG mobility. For example, in Lake Erken, virulent Uroviricota phages carried the glycopeptide resistance gene *vanY*, likely acquired from *Polynucleobacter* hosts. This gene is known for contributing resistance by altering the target of glycopeptides, such as cell wall precursors, allowing bacteria to survive in the presence of antibiotic (e.g., vancomycin). Whether this phage–bacterium pair reflects the legacy of a transfer in the past, or inactive host targeting at the time of sampling, thereby maintaining an ARG reservoir, remains unresolved. Nevertheless, our results add to emerging evidence that viral dynamics can contribute to ARG carriage but that their role in vertical dissemination across stratified layers is greatly limited, indicated by the low level of MAG-ARGs mobilised by plasmids and viruses. Although mobilization of ARGs has been reported at higher frequencies in wastewater treatment plants[25], our finding is consistent with studies systematically investigating ARG mobility, which suggests that mobility depends on taxonomy[26] and resistance mechanism[27].

**Fig. 5 | Predicted lifestyle of ARG-carrying phages (≥ 3 kbp) in freshwaters.** ARGs were conferring resistance either to multiple antibiotics (i.e., phenicol, fluoroquinolone, diaminopyrimidine as shown by the overlapping symbols) or to glycopeptides. For details on Sample locations, see Supplementary data 16, 17.

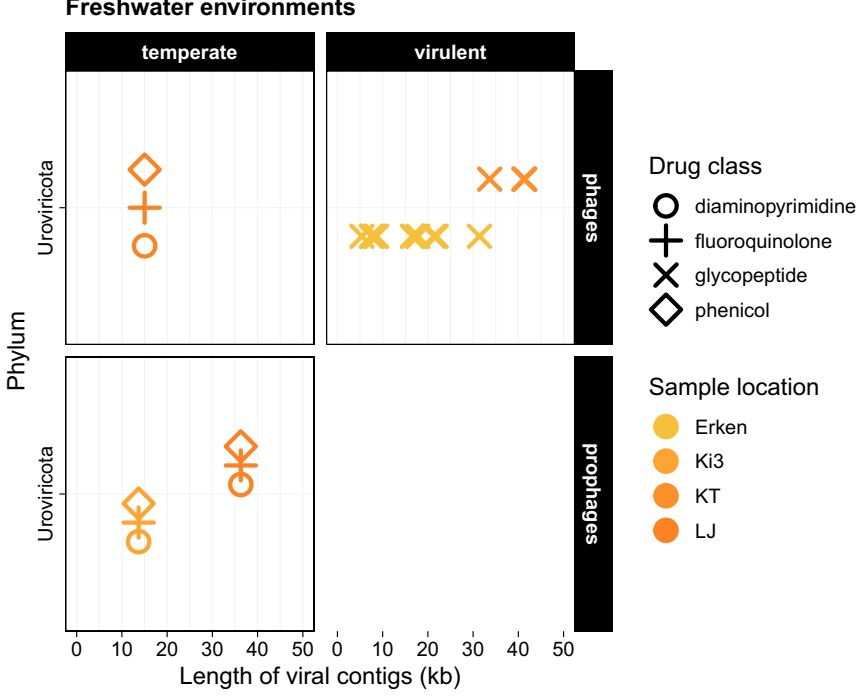

The most widely distributed ARGs appear to be maintained primarily through vertical inheritance rather than recent horizontal transfer. Although a few ARGs were located on MGEs, thus, mobilized (Freshwater: 6.25%, Marine: 14.7%), the majority of ARGs were found across multiple taxonomic classes without evidence of recent HGT events. Examples from the sampled Canadian ponds support this interpretation. Here, Chlorobia bacteria in bottom layers acted as frequent HGT donors to diverse bacteria in upper layers, yet their ARGs (i.e., *vanT* [*vanG* cluster], *vanG*, and *qacG*) showed no evidence of recent horizontal transfer. Instead, these genes were shared across multiple taxa, consistent with vertical inheritance and selective maintenance rather than evidence of recent horizontal dissemination. Notably, the most common ARGs detected confer resistance mechanisms (e.g., target alteration) that are inherently less prone to mobilization, as suggested by Nielsen et al.[27]. Together, these results suggest that while stratified waters do not limit functional gene transfers, ARGs remain largely constrained to lineage-specific inheritance.

One possible explanation for the absence of evident vertical HGT-mediated ARG transfer is unequal spatial distribution of antibiotics and other potential co-selecting agents at different water depths[13], that would otherwise select for resistance genes more evenly across depths. Nonetheless, due to the lack of measurements of antibiotic concentrations, we cannot support or rule out this explanation. Another explanation may be the genetic incompatibility between donor and recipient cells that can reduce the likelihood of ARG exchanges between phylogenetically distinct bacteria[8]. However, the majority of metagenome-assembled genomes analysed here exhibited high sequence similarity (i.e., average nucleotide similarity ≥ 80%; Freshwater: 82.6%, Marine: 62.2%) (Supplementary data 14, 15 and Supplementary Fig. 12), within the range where HGT can occur[15]. Thus, genetic incompatibility is unlikely to explain the lack of HGT-mediated ARG dissemination across layers. Instead, methodological constraints may potentially play a role. Specifically, metagenomic assembly tools often fail to recover intact ARGs, especially those existing in multiple genetic contexts[28], and at the same time tools such as MetaCHIP detect recent transfers but may miss older or highly divergent events[15]. While these caveats underscore the need for methodological advances, they do not diminish our conclusion that ARG dissemination across water layers is subtle in stratified waters.

As most of the lakes analysed in this study are located in Northern Europe and Canada, the microbial community composition and ARG distribution patterns observed here may not fully represent oxygen-stratified freshwater ecosystems globally. Environmental factors such as climate, trophic status, land use, and regional anthropogenic pressures vary substantially across geographic regions and may influence both microbial assemblages and the dynamics of horizontal gene transfer. In addition, the number of samples was uneven across freshwater habitats, which may bias the representation of certain lake types and limit direct extrapolation of quantitative patterns. Consequently, while the observed trends are robust within the studied systems, caution is warranted when generalizing these findings to stratified freshwater ecosystems in other geographical contexts. Furthermore, although our analysis primarily contrasts freshwater and marine ecosystems, we acknowledge that environmental and geographic factors, such as nutrient load, water flow, and local physicochemical conditions likely influence microbial functions, interactions, and the overall ARG dynamics, which potentially provide an explanation for the lack of HGT events between communities at different depths in lake Le Loclat and in several sampling sites scattered in the Southern Hemisphere. Due to limited and inconsistent availability of these variables across datasets, a quantitative assessment of their relative contributions was not possible. Our results, stratified by ecosystem type and sampling depth, provide a baseline for comparative HGT and ARG patterns, while future studies incorporating detailed environmental metadata will be essential to fully resolve the impact of ecological factors.

Nonetheless, our findings provide a more comprehensive evaluation of ARG distribution than prior works, linking water stratification and limited vertical HGT to the persistence and structuring of resistance genes across water columns. Our study has important implications for understanding the ecology of antimicrobial resistance. In contrast to mixed water columns that promotes cell-to-cell interactions and hence genetic exchange[1], stratification appears to act as a natural barrier against vertical spread of ARGs, likely in part by influencing the distribution of stressors at varying extent across water column. Climate change is projected to intensify and prolong stratification in both marine[9] and freshwater ecosystems[11]. Our findings, therefore, point to further limitation in the vertical dissemination of resistance genes by stratification, while

potentially fostering within-layer accumulation. In conclusion, aquatic environments are less likely to spread clinically relevant ARGs between known bacterial hosts, however, yet-to-be-discovered resistance determinants and the many uncultured species present in these environments might underestimate the importance of such environments[8]. Future work should combine genome-centric methods with explicit tracking of ARG-carrying plasmid and viruses across stratification cycles to better predict the fate of resistance genes.

## Methods

### Data selection and processing

To assess putative gene transfers across water depths, we selected publicly available metagenomic data with at least three vertical samplings from freshwater[29] and marine[30] ecosystems (for full sample list, see Supplementary data 16, 17). All freshwater data originated from oxygen-stratified waters; however, we needed to validate whether the sampled marine environments were also stratified. To do so, we gathered hydrological data from GEOTRACES webODV website (geotraces.webodv.awi.de) and used a density-based method for defining mixed-layer depth, as described in ref. 10. Except one sampling location (i.e., GP13_2), due to missing sensor data, all sites showed stratification with varying mixed-layer depths (in a range of 9–110.1 m, median: 16.2 m; Supplementary data 18). The sampling design of both datasets, i.e., metagenomic data spanning across both the mixed layer and below, enabled us to assess horizontal gene transfers across stratified water layers.

Sequencing data was processed using nf-core/mag v2.5.1[31] of the nf-core collection of workflows[32] executed in Nextflow v23.10.0[33]. Briefly, raw reads were pre-processed by adapter and quality trimming with fastp (v0.23.4), removing PhiX reads with Bowtie2 (v2.4.2), and quality controlled with FastQC (v0.11.9). The pre-processed reads of each sample were then assembled individually using MEGAHIT (v1.2.9), and the resulting contigs were binned into MAGs by MetaBAT2 based on nucleotide frequencies and co-abundance patterns across samples from the same vertical profile. MAG abundances were estimated by default for the different samples from contig sequencing depths. Quality control features of the generated assemblies and MAGs were assessed by QUAST and MAG completeness and contamination was estimated by BUSCO and taxonomically classified by GTDB-Tk[34]. A multiQC generated comprehensive summary of the executed pipeline, including quality control metrics and software versioning, have been deposited on the project's GitHub repository (see Code Availability).

### Mobile genetic elements

MGEs such as plasmids and viruses (with ≥ 3kbp contigs) were identified in each assembly by geNomad (v1.5.2)[35] with default settings. The viral classified elements can originate from phages that infect bacteria and have two distinct lifestyles: virulent and temperate. To predict the lifestyle of phages in our samples (PhaTYPScore ≥ 0.75), we used the PhaTYP module[19] within PhaBOX[20] to reveal potential phage-mediated processes in ARG dissemination.

### Predicting community-level horizontal gene transfers

The resulting metagenomic bins with ≥ 40% completeness served as input for subsequent analyses to detect putative HGT events across vertical sampling depths using the MetaCHIP pipeline[15]. Such putatively transferred genes are identified as those parts of assemblies showing high-homology alignments to alternative genomes and little homology to the flanking regions[2]. To get insights into the functional roles of HGT-involved genes, we used DeepNOG[36] with the eggNOG5 reference database[37] to annotate genes to functions.

To describe the frequency of HGT events across depths in each sampling location, we introduce *HGT activity* which refers to the proportion of HGT-mediated unique genes, expressed as a percentage of the total number of ORFs within a sampling location (Eq. 1). This scaling approach enabled to compare the HGT activities across locations and environments by using Wilcoxon rank-sum test and to calculate its effect size (r). Rarefaction curves

of ORFs demonstrated that HGT activity rapidly stabilized with increasing sampling effort, indicating that this metric is robust to differences in sequencing depth and assembly size (Supplementary Fig. 13)

$$HGT\ activity(\%) = \frac{Number\ of\ HGT - mediated\ unique\ genes}{Total\ number\ of\ predicted\ genes} \times 100$$

(1)

To identify differences in donor versus recipient roles across taxonomic groups (i.e., classes), Fisher's exact test was performed with Monte Carlo simulations (*B* = 999) to account for sparse counts. To test whether HGT-mediated genes were evenly distributed across COG categories, chi-squared goodness-of-fit tests were done against a uniform expectation, and standardized residuals were calculated to identify the most significantly enriched category.

### Identification of antibiotic resistance genes

Since environmental stressors can select for and maintain ARGs, we first assessed the frequency of genes related to various defence mechanisms. For this, ORFs of MAGs were predicted by Prodigal (v2.6.3)[38] and those associated with defence mechanisms (COG category 'V'), as proxy for the presence of stressors, were identified by using DeepNOG[36] with the egg-NOG5 reference database[37]. This step confirmed that genomes, particularly from lakes, ponds and the Atlantic Ocean, were exposed to stressors, potentially paving the way for the selection and/or the maintenance of ARGs (Supplementary Fig. 5). As a next step, both predicted genes of MAGs and MGEs were mapped against three distinct ARG databases to comprehensively assess the presence of established ARGs from both pathogenic and non-pathogenic bacteria, as well as those that are computationally predicted (which are largely latent). Specifically, all genes were: (i) annotated using the RGI tool against The Comprehensive Antibiotic Resistance Database (CARD)[16] under the strict paradigm option, (ii) mapped ( ≥ 90% identity and ≥ 20% coverage) using BLASTp (v2.14.1)[39] against the ResFinderFG2.0 database[18] to target latent ARGs present in non-culturable and/or non-pathogenic bacteria, as well as (iii) against a database of computationally predicted latent ARGs[17] to detect less characterized ARGs. The relationship between ARG richness and water depth was evaluated with Spearman correlation test.

Similarly, HGT-involved genes were also mapped to these three databases to identify ARGs that are disseminated by HGT across water depths.

### Linking mobile genetic elements to MAG-ARGs

Using strict thresholds ( ≥ 95% identity and ≥ 70% coverage), we mapped resistance genes found in plasmid/virus contigs (MGE-ARGs) to ARGs in genome bins (MAG-ARGs) by using BLASTp (v2.14.1)[39]. This step allowed the assessment of ARG mobility and the detection of potential phage-mediated spread of ARGs across different water depths.

Custom Python scripts were written to process data and perform data analyses using the above-mentioned tools (see our flowchart in Supplementary Fig. 1), and the packages of ggOceanMaps[40] and ggplot2[41] were used in R (v4.4.3)R Core[42] to visualise the results. All scripts have been reposited in the project's GitHub repository (see Code Availability).

### Reporting summary

Further information on research design is available in the Nature Portfolio Reporting Summary linked to this article.

### Data availability

Metagenomic data used in this study was obtained from publicly available data collections: Buck et al. (2021) and Biller et al. (2018) for samples from freshwater and marine environments, respectively. All sequences are deposited to the European Nucleotide Archive under the project number PRJEB38681[43] and PRJNA385854[44]. Oceanographic data for stratification

assessment was extracted from the GEOTRACES webODV website (geo-traces.webodv.awi.de). Source data underlying the figures can be found in supplementary data 1-5.

## Code availability

Sequencing data processing pipeline summary files, including quality reports, as well as all python and R scripts used in this study have been deposited in: github.com/mtva0001/BlendARGs and also accessible on Figshare with the following https://doi.org/10.6084/m9.figshare.31353151[45].

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

## Acknowledgements

This study was funded by the Data-Driven Life Science (DDLS) program supported by the Knut and Alice Wallenberg Foundation (KAW 2020.0239). J.B.P. acknowledges funding from the Swedish Research Council (VR; grants 2019-00299 and 2023-01721) under the frame of JPI AMR (EMBARK and SEARCHER; JPIAMR2019-109 and JPIAMR2023-DISTOMOS-016, respectively), and the Swedish Foundation for Strategic Research (FFL21-0174), M.V. was supported by the Swedish Research Council (VR Starting grant, no. 2024-05922). We thank to Buck et al. (2021), Biller et al. (2018) and the GEOTRACES programme for their open-source datasets that made this work possible. The computations and data handling were enabled by resources provided by the National Academic Infrastructure for Super-computing in Sweden (NAISS), funded by the Swedish Research Council (2022-06725).

## Author contributions

Máté Vass: Conceptualization, Formal analysis, Visualization, Data curation, Writing – original draft, review. Anna Abramova: Methodology, Investigation, Writing–review, and editing. Johan Bengtsson-Palme: Conceptualization, Methodology, Writing – review and editing.

## Funding

## Competing interests

The authors declare no competing interests.
