## [Transparent Peer Review file · Communications Biology]

Antimicrobial resistance dissemination via horizontal gene transfer is constrained in stratified waters

Corresponding Author: Dr Máté Vass

Version 0:

Reviewer comments:

Reviewer #1

(Remarks to the Author)

This manuscript presents comparative analysis of horizontal gene transfer (HGT) and antibiotic resistance gene (ARG) distribution across freshwater and marine water columns. While the work addresses important questions about microbial gene mobility in aquatic ecosystems, several methodological concerns and interpretive issues require attention.

Lines 87-89 "4,562 HGT events" in freshwater vs. "100 in marine" represents a 45-fold difference that seems biologically implausible without considering confounding factors

MetaCHIP predictions are computational estimates prone to false positives, especially in complex metagenomes, no validation through GC content/codon usage bias examination. Please discuss false positive rates explicitly.

Line 91 "Le Loclat" and Southern Hemisphere marine sites had zero HGT—why? Data quality issue or biological?

Lines 96-102 Effect size interpretation should be revised; $r = 0.48$ described as "moderate effect" but $p < 0.001$ with $n=68$ total—effect could be driven by outliers. Please show distribution plots.

Line 98-99 "No significant differences after correction" - which correction? Bonferroni? FDR?

Lines 101-102 Correctly note small sample sizes but still report effect sizes—these should not be reported or should include massive confidence intervals. Show raw data distributions (boxplots, violin plots), report confidence intervals for all effect sizes, consider Bayesian approaches given small samples.

Lines 103-106 Dividing by total ORFs assumes equal sequencing depth and assembly quality across environments—unlikely given the stated depth sampling differences. Also, if marine samples have fewer MAGs (1,489 vs 7,865), the denominator is smaller, potentially inflating marine HGT activity contrary to the findings. Wilcoxon test assumes independent samples, but water column depths within sites are spatially autocorrelated. Please justify why this metric is superior to alternatives (e.g., HGT events per MAG, per taxon) add control for sequencing depth, assembly quality (N50, completeness), provide rarefaction curves showing metric stability.

Lines 103-106 The authors acknowledge that: freshwater - narrow depth layers and marine- large depth intervals. This is a confounding variable that could entirely explain the freshwater vs. marine difference: Finer sampling resolution in freshwater = more similar microbial communities = more detectable recent HGT. Coarser marine sampling = communities separated by greater ecological/temporal distance = fewer detectable transfers

The Supplementary Fig. 2 reference is insufficient - this should be:

1. Quantitatively analyzed (correlation between depth interval and HGT activity)
2. Statistically controlled (e.g., ANCOVA with depth interval as covariate)

Lines 174-175 "Surprisingly, none of the identified ARGs were involved in the estimated HGT events" This is the paper's most striking finding but is under-discussed: MetaCHIP may not detect ARG transfers (parameter tuning issue?) HGT events are ancient (pre-stratification), so depth-based analysis misses them. With only 215 total ARGs and 4,662 HGT events, overlap probability is low by chance.

What percentage of non-ARG genes were involved in HGT? (baseline comparison needed) Are ARGs on mobile elements

(plasmids, integrons) excluded from MAG-based HGT detection? Were ARGs excluded due to quality filters in MetaCHIP? Please discuss whether vertical inheritance dominates ARG distribution in these environments.

Lines 176-186 Why is glycopeptide resistance dominant in both environments? (Vancomycin is not an environmental antibiotic)

Lines 228-247 The MGE analysis partially contradicts the HGT analysis: 6.25-14.71% of ARGs found on MGEs (plasmids/phages), Lake Erken shows phage-mediated vanY dissemination (lines 242-245). Marine plasmids show "slightly greater dissemination" (lines 246-247)

1.If ARGs are on MGEs, why does MetaCHIP detect zero HGT events involving ARGs?

2.The 1:1 line interpretation (lines 238-241) is unclear - does this mean no vertical movement or that MGEs and MAGs co-occur at the same depth?

Please reconcile MetaCHIP (zero ARG-HGT) with MGE findings (some ARG mobility).

Clarify Figure 4c,f interpretation - provide statistical test for deviation from 1:1 line.

Discuss why only 6-15% of ARGs are mobilized vs. literature reports (often >30%)

Table 1: Shows zero HGT-mediated ARGs but doesn't show total HGT events for comparison.

Figure 3: Difficult to interpret - consider different colors.

This manuscript tackles an important question with a substantial dataset, but methodological limitations and statistical issues undermine confidence in the primary conclusion that freshwater environments have higher HGT activity than marine systems. The sampling bias (depth intervals) is a critical confound that must be addressed. The finding of zero HGT-mediated ARG transfers is striking but requires deeper investigation to determine if it's methodological artifact or biological reality.

Reviewer #2

(Remarks to the Author)

This study presents an interesting analysis of depth-resolved metagenomic data from stratified freshwater and marine environments to evaluate the role of HGT in the spread of ARGs. However, the analytical depth and overall presentation require improvement to fully support the authors' conclusions. The specific points are detailed below.

The study does not sufficiently address how the distinct ecological settings (freshwater vs. marine), geographic location, and key physiochemical (e.g., nutrient load, water flow) influence biological phenomenon, such as microbial functions and interactions. A comparative analysis of the relative importance of these factors on ARG abundance and HGT is needed.

The geographical distribution of samples, particularly for freshwater environments, is highly uneven. The authors should address the potential biases this may introduce and how they might affect the generalizability of the findings.

The claim of demonstrating "HGT across depths" (Figure 1 and main text) appears overstated. The MetaCHIP tool identifies potential HGT events within a community but does not directly confirm contemporary cross-depth transfer.

Essential metadata, such as sequencing depth, sampling time, and detailed geographic distribution of samples, are inadequately described. Given the use of public data, a clear description of the data sources and quality control measures is crucial.

The definitions of some important terms in the manuscript are unclear. For example, MGE-ARG was defined in L230-234 ".....the predicted genes of contigs classified as MGEs were mapped to ARG-carrying bacterial genomes (hereafter MAG-ARGs). We assumed that a successful match would indicate that the resistance gene is part of the mobilome (MGE-ARGs) and can potentially be spread spatially by plasmids and viruses." The inference of an MGE-ARG association appears to be based on their co-occurrence within a single genome. This approach is ambitious, as it does not demonstrate a direct physical linkage on a mobile genetic element.

Terminology and Presentation:

The use of overlapping terms (e.g., "HGT-mediated ARGs" and "MGE-ARGs") is confusing and should be clarified and standardized.

Abbreviations are re-introduced multiple times; a single definition upon first use would improve readability.

The text in Figure 5 and Table 1 is blurry and difficult to read.

Table 1 contains minimal information and could be removed.

Figures 2 & 3: The color schemes are difficult to distinguish. The methodological descriptions in the figure legends should be moved to the main Methods section.

In Figure 4, panels (b) and (e) show limited variation (mostly one unique ARG) and may not be necessary. The meaning of

the vertical axis in this figure is unclear, as the boxes appear to represent categories rather than numerical values.

Version 1:

Reviewer comments:

Reviewer #1

(Remarks to the Author)

Reviewer #2

(Remarks to the Author)

All concerns and questions have been properly addressed.

Rebuttal Letter

We appreciate the time that the reviewers dedicated to providing feedback on our study. We have incorporated several suggestions made by the reviewers and highlighted them within the tracked version of the revised manuscript.

Please see below, in blue, for a point-by-point response to their comments. The line numbers refer to the revised article file.

Reviewers' comments:

Reviewer #1 (Remarks to the Author):

This manuscript presents comparative analysis of horizontal gene transfer (HGT) and antibiotic resistance gene (ARG) distribution across freshwater and marine water columns. While the work addresses important questions about microbial gene mobility in aquatic ecosystems, several methodological concerns and interpretive issues require attention.

Lines 87-89 "4,562 HGT events" in freshwater vs. "100 in marine" represents a 45-fold difference that seems biologically implausible without considering confounding factors. MetaCHIP predictions are computational estimates prone to false positives, especially in complex metagenomes, no validation through GC content/codon usage bias examination. Please discuss false positive rates explicitly.

We appreciate the reviewer's insightful comment. We agree that the large difference in the number of inferred HGT events between freshwater and marine environments warrants clarification. Several underlying factors might have contributed to this observed pattern:

1. As we mentioned in our manuscript, the depth intervals differed greatly between the freshwater and marine datasets. This difference may directly influence the number of detectable HGT events between depths, because we used MetaCHIP to identify transfers between depth-defined communities.
2. Marine environments, especially those away from coastal regions, have substantially lower microbial biomass and nutrient availability compared with freshwaters. Thus, lower biomass reduces both the likelihood for HGT events, as well as the probability of capturing HGTs in MAGs. Thus, biological differences between environments can naturally produce strong asymmetries in observed HGT counts, especially when assessed along water columns. This was the main reason we decided to normalize our estimates by calculating *HGT activity* rather than statistically comparing absolute HGT counts.
3. We agree that computational HGT detection can produce false positives. Importantly, however, the MetaCHIP tool includes multiple filters to minimize this issue.
 - a. During its first step, 'best-match approach', if a sequence has only a self-match in its own group, all cross-group BLASTn matches are discarded. This prevents spurious assignments where a non-self match with high identity could be incorrectly labeled as an HGT.

- b. MetaCHIP also explicitly identifies and removes candidate HGTs located near contig ends with >95% identity, which often arise from assembly “bubbles” produced by several assembly methods.
- c. Putative HGTs on contigs that share >95% full-length similarity with a longer contig are excluded, as these likely represent artificial duplicates rather than gene transfers.
- d. During the phylogenetic approach (2nd step of the MetaCHIP tool), MetaCHIP recommends using genomes with at least 40% completeness to reduce risk rooted in mis-binned sequences that could cause false positives. We followed this recommendation strictly.

These steps collectively remove the majority of systematic errors that would otherwise inflate our HGT estimates.

It is important to emphasize that our study focuses on relative comparisons between environments rather than absolute counts. Because all samples were processed with the same workflow, including bin-quality thresholds, we believe that our comparisons, based on HGT activity rather than absolute HGT counts, remain valid even if absolute HGT counts contain some degree of uncertainty. As the MetaCHIP authors state in their paper: “...despite these limitations, simulated and real data show that HGTs can be detected with a high degree of confidence. However, absolute numbers may be underestimated.” We acknowledge the limitations of purely computational HGT estimation in our Discussion and also added now a paragraph where we highlight this 45-fold difference in HGT activity in L253-260:

We observed substantially higher HGT activity in freshwaters compared with marine sites. The difference is biologically plausible: marine environments, particularly oligotrophic waters, have lower microbial biomass and nutrient availability, which likely reduces both the frequency of HGT events and the probability of detecting them. Sampling resolution along depth may also influence our estimates, with freshwater sites having higher coverage across multiple depth intervals. Therefore, while absolute HGT counts should be interpreted cautiously, the observed pattern reliably reflects higher relative HGT activity in freshwater communities.

Line 91 "Le Loclat" and Southern Hemisphere marine sites had zero HGT—why? Data quality issue or biological?

We were also puzzled to explain the lack of detectable HGT events between communities at different depths at these sites. We could not find issues in the sequencing data quality (MultiQC reports are available on the project’s github page) and even after MetaCHIP analysis was re-run to double-check the results, we couldn’t find a clear reasoning for this pattern. Nevertheless, we now added a new paragraph for the Discussion (L339-358) to emphasize that local environmental factors might have influenced microbial communities which in turn resulted in the observed patterns:

As most of the lakes analysed in this study are located in Northern Europe and Canada, the microbial community composition and ARG distribution patterns observed here may not fully represent oxygen-stratified freshwater ecosystems globally. Environmental factors such as climate, trophic status, land use, and regional anthropogenic pressures vary substantially across geographic regions and may influence both microbial assemblages and the dynamics of horizontal gene transfer. In addition, the number of samples was uneven across freshwater

habitats, which may bias the representation of certain lake types and limit direct extrapolation of quantitative patterns. Consequently, while the observed trends are robust within the studied systems, caution is warranted when generalizing these findings to stratified freshwater ecosystems in other geographical contexts. Furthermore, although our analysis primarily contrasts freshwater and marine ecosystems, we acknowledge that environmental and geographic factors, such as nutrient load, water flow, and local physicochemical conditions likely influence microbial functions, interactions, and the overall ARG dynamics, which potentially provide an explanation for the lack of HGT events between communities at different depths in lake Le Loclat and in several sampling sites scattered in the Southern Hemisphere. Due to limited and inconsistent availability of these variables across datasets, a quantitative assessment of their relative contributions was not possible. Our results, stratified by ecosystem type and sampling depth, provide a baseline for comparative HGT and ARG patterns, while future studies incorporating detailed environmental metadata will be essential to fully resolve the impact of ecological factors.

Lines 96-102 Effect size interpretation should be revised; $r = 0.48$ described as "moderate effect" but $p < 0.001$ with $n=68$ total—effect could be driven by outliers. Please show distribution plots.

We have rephrased the interpretation of the effect size. The value of $r=0.48$ falls within the range of “moderate effect” by `wilcox_effsize` function in the `rstatix` R package. Based on their description, “the interpretation values for r commonly in published literature and on the internet are: $0.10 - < 0.3$ (small effect), $0.30 - < 0.5$ (moderate effect) and ≥ 0.5 (large effect)”. Nevertheless, we now note that this effect size should be interpreted cautiously due to non-normal data and potential influence of outliers. To illustrate the data distribution, we added a violin plot to the Supplementary Fig. 2:

This visualization justifies the use of Wilcoxon rank-sum test, which was chosen after Shapiro-Wilk test confirmed non-normality.

The rephrased text (L94-99):

We observed significantly greater HGT activity in freshwater than in marine communities (Wilcoxon rank-sum test, $p < 0.001$, $r = 0.48$) (Fig. 1). While this effect size is commonly interpreted as a moderate effect, we caution that this estimate may be influenced by the non-normal distribution of the data (Supplementary Fig. 2). Therefore, we rely primarily on the direction of the median difference rather than the conventional r thresholds alone.

Line 98-99 "No significant differences after correction" - which correction? Bonferroni? FDR?

We now specify the applied Holm-Bonferroni method in L99-103:

Pairwise comparisons of HGT activity within freshwater environments revealed no statistically significant differences after Holm-Bonferroni correction for multiple testing (all adjusted $p > 0.05$), however, this should be interpreted with caution due to the small sample sizes in pond ($n = 5$) and reservoir ($n = 1$) categories.

Lines 101-102 Correctly note small sample sizes but still report effect sizes—these should not be reported or should include massive confidence intervals. Show raw data distributions (boxplots, violin plots), report confidence intervals for all effect sizes, consider Bayesian approaches given small samples.

We thank for pointing this out. We removed the estimates and rephrased this section (L99-103):

Pairwise comparisons of HGT activity within freshwater environments revealed no statistically significant differences after Holm-Bonferroni correction for multiple testing (all adjusted $p > 0.05$), however, this should be interpreted with caution due to the small sample sizes in pond (n

= 5) and reservoir (n = 1) categories.

Lines 103-106 Dividing by total ORFs assumes equal sequencing depth and assembly quality across environments—unlikely given the stated depth sampling differences. Also, if marine samples have fewer MAGs (1,489 vs 7,865), the denominator is smaller, potentially inflating marine HGT activity contrary to the findings. Wilcoxon test assumes independent samples, but water column depths within sites are spatially autocorrelated. Please justify why this metric is superior to alternatives (e.g., HGT events per MAG, per taxon) add control for sequencing depth, assembly quality (N50, completeness), provide rarefaction curves showing metric stability.

To compare the frequency of HGT across sampling locations/environments, we quantified HGT activity as the proportion of HGT-mediated unique genes relative to the total number of ORFs per sampling sites. This normalization does not assume equal sequencing depth or assembly quality (though we used MAGs with $\geq 40\%$ completeness as recommended to reach reliable estimates), rather, it scales HGT counts to the effective gene space recovered at each location, thereby reducing sensitivity to differences in sequencing effort, and MAG number and bin sizes due to varying species diversity and biomass.

Metrics such as HGT events per MAG or per taxon would disproportionately weight sites with more uneven taxonomic representation. Using the proportion of HGT-mediated genes, therefore, provides a more consistent unit of comparison across heterogeneous metagenomic datasets.

We acknowledge that water depths within sites exhibit spatial autocorrelation, however, our analysis is conducted at the level of independent sampling locations across environments, each represented by a depth-integrated measure of HGT activity. While differences in sequencing depth and assembly quality can influence ORF recovery, this effect is implicitly controlled by scaling to the total ORF count.

The requested rarefaction plot confirms that the curves quickly stabilize, indicating that our metric is robust:

We include this plot as Supplementary Fig. 13 and added the following sentence to the main text in L429-432:

Rarefaction curves of ORFs demonstrated that HGT activity rapidly stabilized with increasing sampling effort, indicating that this metric is robust to differences in sequencing depth and assembly size (Supplementary Fig. 13).

Lines 103-106 The authors acknowledge that: freshwater - narrow depth layers and marine- large depth intervals. This is a confounding variable that could entirely explain the freshwater vs. marine difference: Finer sampling resolution in freshwater = more similar microbial communities = more detectable recent HGT. Coarser marine sampling = communities separated by greater ecological/temporal distance = fewer detectable transfers

The Supplementary Fig. 2 reference is insufficient - this should be:

1. Quantitatively analyzed (correlation between depth interval and HGT activity)
2. Statistically controlled (e.g., ANCOVA with depth interval as covariate)

We appreciate the statistical suggestions and for highlighting depth interval differences as a potential confounding factor.

Quantitative analysis showed indeed a significant negative correlation between median depth interval and HGT activity (Spearman rho=-0.59, p<0.001). ANCOVA test including median depth interval as a covariate confirmed that the effect of environment (Freshwater vs. Marine; labelled as ‘Source’ in our ANCOVA model) remained significant (p=0.021), while the covariate and interaction were not significant, see summary table:

Call:

lm(formula = HGT_activity ~ Source * median_distance, data = combined_median_distances)

Residuals:					
	Min	1Q	Median	3Q	Max
	-0.07135	-0.013656	-0.003888	0.008849	0.211733
Coefficients:					
	Estimate	Std. Error	t value	Pr(> t)	
(Intercept)	0.087266	0.015141	5.763	1.44E-06	***
SourceM	-0.074876	0.031091	-2.408	0.0213	*
median_distance	-0.008616	0.007368	-1.169	0.2499	
SourceM:median_distance	0.008642	0.007375	1.172	0.249	

Residual standard error: 0.0479 on 36 degrees of freedom

Multiple R-squared: 0.3241, Adjusted R-squared: 0.2677

F-statistic: 5.753 on 3 and 36 DF, p-value: 0.002539

Our conclusion, therefore, is that although coarser marine sampling reduced HGT detectability to some extent, the freshwater-marine difference cannot be fully attributed to sampling resolution.

We complemented our results in the main text with the following in L104-115:

Vertical sampling differed in depth resolution; in the shallow freshwater environments, samples were taken from narrower layers, whereas in marine environments the sampled layers spanned over larger depth intervals. This may have influenced the estimated HGT activity due to the larger depth intervals in marine environments (Supplementary Fig. 4). We observed a negative correlation between median depth interval and HGT activity (Spearman rho = -0.59, $p < 0.001$), suggesting that coarser vertical sampling in marine environments reduced the detectability of HGT events. To statistically control for this confounding factor, we performed an ANCOVA including median depth interval as a covariate and testing for interaction with environment. Environment remained a significant predictor of HGT activity ($p = 0.021$), while depth interval and the interaction were not significant ($p = 0.25$), indicating that the observed environmental difference in HGT activity is not fully explained by differences in sampling resolution.

Additionally, we updated the corresponding figure in the Supplementary:

Supplementary Figure 4. Median depth interval (m) in each sampling location in the presence of horizontal gene transfer (HGT). HGT activity refers to the proportion of HGT-mediated unique genes, expressed as a percentage of the total number of genes (ORFs) within a sampling location. Black lines refer to regression lines per environment with standard error.

Lines 174-175 "Surprisingly, none of the identified ARGs were involved in the estimated HGT

events" This is the paper's most striking finding but is under-discussed: MetaCHIP may not detect ARG transfers (parameter tuning issue?) HGT events are ancient (pre-stratification), so depth-based analysis misses them. With only 215 total ARGs and 4,662 HGT events, overlap probability is low by chance.

What percentage of non-ARG genes were involved in HGT? (baseline comparison needed) Are ARGs on mobile elements (plasmids, integrons) excluded from MAG-based HGT detection? Were ARGs excluded due to quality filters in MetaCHIP?

We agree that MetaCHIP could potentially miss some HGT events, leading to underestimated HGT activity. We have already discussed potential limitations of this approach, for instance in L331-3337:

Instead, methodological constraints may potentially play a role. Specifically, metagenomic assembly tools often fail to recover intact ARGs, especially those existing in multiple genetic contexts (Abramova et al., 2024), and at the same time tools such as MetaCHIP detect recent transfers but may miss older or highly divergent events (Song et al., 2019). While these caveats underscore the need for methodological advances, they do not diminish our conclusion that ARG dissemination across water layers is subtle in stratified waters.

Since no HGT-mediated ARG dissemination was found, all reported HGT activity (%) refers to non-ARG genes involvement in HGT, hence, can be seen as baseline.

MGE contigs can end up in MAGs and we did not conduct any filtration on such contigs in MAGs. ARGs were present in these MAGs (as we show in Fig. 3). Any potential ARG removal by MetaCHIP might have occurred when partial ARGs were present at the end of contigs (common issue in genome assemblies) and MetaCHIP filters them out to mitigate false positive candidates because of the concern of assembly “bubbles”, discussed in our previous response above.

Please discuss whether vertical inheritance dominates ARG distribution in these environments.

We extended the current paragraph in the main text (L306-318) to emphasize this observation: *The most widely distributed ARGs appear to be maintained primarily through vertical inheritance rather than recent horizontal transfer. Although a few ARGs were located on MGEs, thus, mobilized (Freshwater: 6.25 %, Marine: 14.7 %), the majority of ARGs were found across multiple taxonomic classes without evidence of recent HGT events. Examples from the sampled Canadian ponds support this interpretation. Here, Chlorobia bacteria in bottom layers acted as frequent HGT donors to diverse bacteria in upper layers, yet their ARGs (i.e., vanT [vanG cluster], vanG, and qacG) showed no evidence of recent horizontal transfer. Instead, these genes were shared across multiple taxa, consistent with vertical inheritance and selective maintenance rather than evidence of recent horizontal dissemination. Together, these results suggest that while stratified waters do not limit functional gene transfers, ARGs remain largely constrained to lineage-specific inheritance and are generally less mobilized as suggested by (Nielsen et al., 2022). Together, these results suggest that while stratified waters do not limit functional gene transfers, ARGs remain largely constrained to lineage-specific inheritance.*

Lines 176-186 Why is glycopeptide resistance dominant in both environments? (Vancomycin is

not an environmental antibiotic)

Vancomycin originates from the soil bacterium *Amycolatopsis orientalis*, as a defense against other bacteria. Beside terrestrial environment, the marine core resistome comprises of a great fraction of glycopeptide resistance genes (Xu et al., 2023. Water Research), even if certain regions show otherwise (i.e., South China Sea; Lu et al., 2024. Frontiers in Environmental Science and Engineering). Vancomycin-resistant ARGs are also common in rivers as summarized by Nnadozie and Odume (2019. Environmental Pollution). Glycopeptide resistance was found as the second most common one in the Baltic Sea (Serrana et al., 2025. Microbiome).

Lines 228-247 The MGE analysis partially contradicts the HGT analysis: 6.25-14.71% of ARGs found on MGEs (plasmids/phages), Lake Erken shows phage-mediated vanY dissemination (lines 242-245). Marine plasmids show "slightly greater dissemination" (lines 246-247)
1.If ARGs are on MGEs, why does MetaCHIP detect zero HGT events involving ARGs?
Please reconcile MetaCHIP (zero ARG-HGT) with MGE findings (some ARG mobility).

We do not see it as a contradiction. There can be several reasons behind this finding. From HGT angle:

- If an ARG is present on an MGE that moves between closely related strains, MetaCHIP may not detect it as HGT candidate because the donor/recipient are too similar.
- If the ARG-carrying contig is not confidently binned into a MAG, MetaCHIP will not include it in the MAG-based HGT estimation.
- Potentially fragmented ARGs sitting near the contig ends can easily be filtered out by MetaCHIP.

From MGE angle:

- We were locating ARG sequences directly on contigs annotated as plasmid/phage. Finding an ARG on an MGE indicates mobility potential but does not guarantee that a detectable HGT between taxonomic units occurred and was captured in MAGs.

Nevertheless, given that numerous non-ARG HGT events were detected, we believe that our chosen method, MetaCHIP, was able to detect HGTs across MAGs. Our results point to an ARG specific phenomenon. The most likely scenario is that only a small subset of ARGs were on MGEs and many ARG transfers are either ancient or vertically maintained after an initial transfer. MetaCHIP looks for relatively recent HGT events in a reliable manner.

Taken together, these two findings are not contradictory at all: one suggests a few ARGs are located on MGEs (=mobility potential), the other suggests we did not detect recent, bin-resolved, phylogenetically supported ARG transfer between MAGs.

2.The 1:1 line interpretation (lines 238-241) is unclear - does this mean no vertical movement or that MGEs and MAGs co-occur at the same depth?

The 1:1 line is a depth-matching reference. It means that a specific MAG-ARG detected at a given depth was found on a phage or plasmid. In other words, a specific MAG-ARG was not disseminated to other depths. Any deviation from this 1:1 line would indicate that the ARG is carried by a MGE that is not found in a MAG at other depths.

Clarify Figure 4c,f interpretation - provide statistical test for deviation from 1:1 line.
 Discuss why only 6-15% of ARGs are mobilized vs. literature reports (often >30%)

We statistically evaluated deviation from the 1:1 line (MGE-ARG depth = MAG-ARG depth) using Wilcoxon signed-rank test as it can handle paired, non-normal data, providing robust results. Overall, plasmids predominantly maintain ARGs within the same depth as their host MAGs, whereas viruses may facilitate limited and irregular vertical dissemination, particularly in freshwater systems. Nevertheless, ARG dissemination across stratified water columns remains generally constrained.

Freshwater:

	Source	n	mean_shift	median_shift	sd	p_value
1	plasmid	45	-0.452	0	0.887	0.000537
2	virus	32	-1.16	-0.55	4.09	0.0787

Due to low sample size of MGE-ARGs in marine systems, we could not infer robust statistical results:

	Source	n	mean_shift	median_shift	sd	p_value
1	plasmid	11	10.5	0	61.1	1
2	virus	1	-62	-62	NA	1

Note: The median shift represents the typical ARG–host depth difference, while the mean shift captures the influence of rarer but larger deviations, explaining why mean and median shift may differ even when the median is centered near zero.

We rephrased the corresponding section in the main text, ensuring to also clarify the 1:1 line interpretation (L209-230):

We also found evidence for a limited vertical distribution of MAG-ARGs by MGE-ARGs in the water columns (Fig. 4c,f). Here, dots along the 1:1 line indicate that MAG-ARGs at a given depth were found in MGEs at the same depth, reflecting limited vertical dissemination of ARGs via plasmids or viruses across the stratified water column. When analysed by MGE type, freshwater plasmid-associated ARGs showed a highly constrained distribution with a small but significant spread towards shallower depths. This was quantified as the paired difference in depth between MGE-ARG occurrence and the reference depth of the MAG-ARG, where negative values indicate shallower positions in the water column (Wilcoxon signed-rank test: median shift: 0, mean shift: -0.45, sd: 0.89, $p < 0.001$). This indicates that while most plasmid-associated ARGs occur at same depths as their hosts, a slight spread (shift) toward shallower depths is present. In contrast, virus-associated ARGs in freshwater exhibited larger variance in depth differences (mean shift: -1.16, sd: 4.09), but this spread was not statistically significant (Wilcoxon signed-rank test: median difference: -0.55, $p = 0.078$). This suggests sporadic vertical displacement events rather than consistent or directional vertical dissemination. This was mainly attributed to Lake Erken, where phages carried the same glycopeptide-resistance vanY (vanB cluster) gene that was present in a Polynucleobacter host (Polynucleobacter sp002292975; MAG: Erken-D6_32_00868) (Fig. 4c, Supplementary Table 8). In marine environments, sample sizes for MGE-ARGs were limited, particularly for viruses, precluding robust statistical inference. However, the median depth difference between MGE-ARG

occurrence and MAG-ARG was centered at zero, indicating no systematic vertical dissemination of marine plasmid-associated ARGs (Fig. 4f).

We now also discuss ARG mobility to a greater extent and provide additional references that support our finding on low ARG mobility (L298-318):

Nevertheless, our results add to emerging evidence that viral dynamics can contribute to ARG carriage but that their role in vertical dissemination across stratified layers is greatly limited, indicated by the low level of MAG-ARGs mobilised by plasmids and viruses. Although mobilization of ARGs has been reported at higher frequencies in wastewater treatment plants (Petrovich et al., 2018), our finding is consistent with studies systematically investigating ARG mobility, which suggest that mobility depends on taxonomy (Ellabaan et al., 2021) and resistance mechanism (Nielsen et al., 2022).

The most widely distributed ARGs appear to be maintained primarily through vertical inheritance rather than recent horizontal transfer. Although a few ARGs were located on MGEs, thus, mobilized (Freshwater: 6.25 %, Marine: 14.7 %), the majority of ARGs were found across multiple taxonomic classes without evidence of recent HGT events. Examples from the sampled Canadian ponds support this interpretation. Here, Chlorobia bacteria in bottom layers acted as frequent HGT donors to diverse bacteria in upper layers, yet their ARGs (i.e., vanT [vanG cluster], vanG, and qacG) showed no evidence of recent horizontal transfer. Instead, these genes were shared across multiple taxa, consistent with vertical inheritance and selective maintenance rather than evidence of recent horizontal dissemination. Notably, the most common ARGs detected confer resistance mechanisms (e.g., target alteration) that are inherently less prone to mobilization, as suggested by (Nielsen et al., 2022). Together, these results suggest that while stratified waters do not limit functional gene transfers, ARGs remain largely constrained to lineage-specific inheritance.

Table 1: Shows zero HGT-mediated ARGs but doesn't show total HGT events for comparison.

We added this information to Table 1 and, as suggested by the reviewer #2, we removed it from the main text but kept it as Supplementary Table S6.

Figure 3: Difficult to interpret - consider different colors.

We updated Figure 2 and 3 by simplifying the plotted data. Instead of class-level taxonomy, we now use phylum-level coloring scheme. In case of Fig. 2, we use a higher-level classification for COG functions, while in Fig. 3, we plot only the ten most common drug classes. We hope these simplified figures help to aid pattern interpretation.

Figure 2. Vertical distribution and feature (i.e., donor or recipient) of horizontally transferred genes, and their assigned phylum-level taxonomy (based on GTDB (Parks et al., 2018)) and functions in various (a) freshwater, such as lake (n=21), pond (n=5) and reservoir (n=1) and (b) marine (Atlantic Ocean: n=36, Pacific Ocean: n=5) environments.

Figure 3. Vertical distribution of the ten most common antibiotic resistance genes and their associated genomes classified at phylum-level taxonomy (based on GTDB (Parks et al., 2018)) in various (a) freshwater, such as lake ($n=21$), pond ($n=5$) and reservoir ($n=1$), and (b) marine (Atlantic Ocean: $n=36$, Pacific Ocean: $n=5$) sites. Comprehensive Antibiotic Resistance

Database (CARD)(Alcock et al., 2023), the ResFinderFG2.0 (Gschwind et al., 2023) and latent ARGs, a database compiled by (Inda-Diaz et al., 2023), were used as reference databases.

This manuscript tackles an important question with a substantial dataset, but methodological limitations and statistical issues undermine confidence in the primary conclusion that freshwater environments have higher HGT activity than marine systems. The sampling bias (depth intervals) is a critical confound that must be addressed. The finding of zero HGT-mediated ARG transfers is striking but requires deeper investigation to determine if it's methodological artifact or biological reality.

Thank you for your feedback! We hope our revised manuscript and the new statistical tests support our conclusions robustly and address your concerns!

Reviewer #2 (Remarks to the Author):

This study presents an interesting analysis of depth-resolved metagenomic data from stratified freshwater and marine environments to evaluate the role of HGT in the spread of ARGs. However, the analytical depth and overall presentation require improvement to fully support the authors' conclusions. The specific points are detailed below.

The study does not sufficiently address how the distinct ecological settings (freshwater vs. marine), geographic location, and key physiochemical (e.g., nutrient load, water flow) influence biological phenomenon, such as microbial functions and interactions. A comparative analysis of the relative importance of these factors on ARG abundance and HGT is needed.

Our study focused on comparing HGT activity and ARG distribution between freshwater and marine ecosystems using available metagenomic datasets. While we include some metadata on geographic location and sampling depth, detailed environmental parameters such as nutrient load, water flow, or other physiochemical variables were not consistently available across all datasets, which limits our ability to systematically and quantitatively assess their relative contributions.

Nevertheless, we accounted for some environmental differences by stratifying our samples by ecosystems (freshwater vs marine) and depth, and by evaluating correlations with sampling depth intervals (newly added results), which serve as proxy for spatial variation. We agree that a systematic analysis incorporating physicochemical parameters would be valuable for assessing the relative importance of environmental factors on ARGs and HGTs, and we now highlight this as a relevant direction for future research.

We added the following paragraph to the Discussions in L339-358:

As most of the lakes analysed in this study are located in Northern Europe and Canada, the microbial community composition and ARG distribution patterns observed here may not fully represent oxygen-stratified freshwater ecosystems globally. Environmental factors such as climate, trophic status, land use, and regional anthropogenic pressures vary substantially across geographic regions and may influence both microbial assemblages and the dynamics of horizontal gene transfer. In addition, the number of samples was uneven across freshwater habitats, which may bias the representation of certain lake types and limit direct extrapolation of

quantitative patterns. Consequently, while the observed trends are robust within the studied systems, caution is warranted when generalizing these findings to stratified freshwater ecosystems in other geographical contexts. Furthermore, although our analysis primarily contrasts freshwater and marine ecosystems, we acknowledge that environmental and geographic factors, such as nutrient load, water flow, and local physicochemical conditions likely influence microbial functions, interactions, and the overall ARG dynamics, which potentially provide an explanation for the lack of HGT events between communities at different depths in lake Le Loclat and in several sampling sites scattered in the Southern Hemisphere. Due to limited and inconsistent availability of these variables across datasets, a quantitative assessment of their relative contributions was not possible. Our results, stratified by ecosystem type and sampling depth, provide a baseline for comparative HGT and ARG patterns, while future studies incorporating detailed environmental metadata will be essential to fully resolve the impact of ecological factors.

The geographical distribution of samples, particularly for freshwater environments, is highly uneven. The authors should address the potential biases this may introduce and how they might affect the generalizability of the findings.

We agree that our chosen dataset is biased and lacks even sample number within freshwater habitats. We now added the following text to the Discussion to highlight this bias (L339-348): *As most of the lakes analyzed in this study are located in Northern Europe and Canada, the microbial community composition and ARG distribution patterns observed here may not fully represent oxygen-stratified freshwater ecosystems globally. Environmental factors such as climate, trophic status, land use, and regional anthropogenic pressures vary substantially across geographic regions and may influence both microbial assemblages and the dynamics of horizontal gene transfer. In addition, the number of samples was uneven across freshwater habitats, which may bias the representation of certain lake types and limit direct extrapolation of quantitative patterns. Consequently, while the observed trends are robust within the studied systems, caution is warranted when generalizing these findings to stratified freshwater ecosystems in other geographical contexts.*

The claim of demonstrating “HGT across depths” (Figure 1 and main text) appears overstated. The MetaCHIP tool identifies potential HGT events within a community but does not directly confirm contemporary cross-depth transfer.

We mention in the Discussion (L334-337) that “[...] tools such as MetaCHIP detect recent transfers but may miss older or highly divergent events (Song et al., 2019). While these caveats underscore the need for methodological advances, they do not diminish our conclusion that ARG dissemination across water layers is subtle in stratified waters.”

We wish to emphasize that MetaCHIP allows users to specify groups (communities) for which HGT events are estimated, thus, all reported HGT event was estimated between communities at different depths and not within each community.

Essential metadata, such as sequencing depth, sampling time, and detailed geographic distribution of samples, are inadequately described. Given the use of public data, a clear

description of the data sources and quality control measures is crucial.

Sequencing data was evaluated by the MultiQC (v1.15) module, as part of the nf-core/mag pipeline. In L404-406, we state: “*A multiQC generated comprehensive summary of the executed pipeline, including quality control metrics and software versioning, have been deposited on the project’s GitHub repository (see Code Availability).*”

Direct link to the reporting summaries: https://github.com/mtva0001/BlendARGs/tree/main/nf-core_pipeline_summary

The geographical information of sampling locations, as well as the sampling dates are listed in Supplementary Table S9 and S10.

The definitions of some important terms in the manuscript are unclear. For example, MGE-ARG was defined in L230-234 “.....the predicted genes of contigs classified as MGEs were mapped to ARG-carrying bacterial genomes (hereafter MAG-ARGs). We assumed that a successful match would indicate that the resistance gene is part of the mobilome (MGE-ARGs) and can potentially be spread spatially by plasmids and viruses.” The inference of an MGE-ARG association appears to be based on their co-occurrence within a single genome. This approach is ambitious, as it does not demonstrate a direct physical linkage on a mobile genetic element.

We thank for raising this concern. We now rephrased the corresponding section in the Results (L190-193) to clarify our definition but also to highlight its limitation to avoid overinterpretation:

[...] we wished to validate our finding from a different perspective and evaluate the presence of ARGs physically encoded on contigs classified as mobile genetic elements (MGE-ARG), independently of MetaCHIP. The presence of such MGE-ARG indicates potential mobility but does not imply recent HGT.

In L200-206:

MGE-ARGs contigs were mapped to ARG-carrying bacterial genomes (hereafter MAG-ARGs) (see Methods for details). We assumed that a successful match would indicate that the same ARG sequence is physically encoded on a plasmid or viral contig and also present in a bacterial genome, suggesting potential mobilization of the MGE-ARG. While this demonstrates physical association of ARGs with MGEs, it does not prove that the MGE and host genome were physically linked within the same cell at the time of sampling, nor does it imply recent HGT.

In the Methods, we also clarify in L467-471:

we mapped resistance genes found in plasmid/virus contigs (MGE-ARGs) to ARGs in genome bins (MAG-ARGs) by using BLASTp (v2.14.1)(Camacho et al., 2009). This step allowed the assessment of ARG mobility and the detection of potential phage-mediated spread of ARGs across different water depths.

Terminology and Presentation:

The use of overlapping terms (e.g., “HGT-mediated ARGs” and “MGE-ARGs”) is confusing and should be clarified and standardized.

We now clarified it in the beginning of the corresponding Result section. See our previous response.

Abbreviations are re-introduced multiple times; a single definition upon first use would improve readability.

We fixed it. However, note that abbreviations are defined in all figure caption, following the journal's guideline.

The text in Figure 5 and Table 1 is blurry and difficult to read.

This is a conversion fault by the submission platform. All figures are also provided as high-quality, individual PDF files.

Table 1 contains minimal information and could be removed.

Given that reviewer #1 suggested the inclusion of total HGT counts in this table, we decided to move it to the Supplementary as Table S6.

Figures 2 & 3: The color schemes are difficult to distinguish. The methodological descriptions in the figure legends should be moved to the main Methods section.

We removed the methodological description from the captions and updated Figure 2 and 3 by simplifying the plotted data. Instead of class-level taxonomy, we now use phylum-level coloring scheme. In case of Fig. 2, we use a higher-level classification for COG functions, while in Fig. 3, we plot only the ten most common drug classes. We hope these simplified figures help to aid pattern interpretation.

Figure 2. Vertical distribution and feature (i.e., donor or recipient) of horizontally transferred genes, and their assigned phylum-level taxonomy (based on GTDB (Parks et al., 2018)) and functions in various (a) freshwater, such as lake ($n=21$), pond ($n=5$) and reservoir ($n=1$) and (b) marine (Atlantic Ocean: $n=36$, Pacific Ocean: $n=5$) environments.

Figure 3. Vertical distribution of the ten most common antibiotic resistance genes and their associated genomes classified at phylum-level taxonomy (based on GTDB (Parks et al., 2018)) in various (a) freshwater, such as lake ($n=21$), pond ($n=5$) and reservoir ($n=1$), and (b) marine (Atlantic Ocean: $n=36$, Pacific Ocean: $n=5$) sites. Comprehensive Antibiotic Resistance

Database (CARD)(Alcock et al., 2023), the ResFinderFG2.0 (Gschwind et al., 2023) and latent ARGs, a database compiled by (Inda-Diaz et al., 2023), were used as reference databases.

In Figure 4, panels (b) and (e) show limited variation (mostly one unique ARG) and may not be necessary. The meaning of the vertical axis in this figure is unclear, as the boxes appear to represent categories rather than numerical values.

As we report in the Results (L207-209) that “ARG-carriage by MGEs was mainly limited to one (or maximum two) unique ARGs per sampling site (Fig. 4b,e), indicating a low level of mobility of ARGs by plasmids or viruses in our selected data”, we wish to keep these panels to directly show the referenced results.

The y-axis in this case refers to the number of ARGs found per sampling site. It shows that there is not a single site where the number of ARGs were enriched in viral or plasmid contigs.